# Method of Monitoring 26S Proteasome in Cells Revealed the Crucial Role of PSMA3 C-Terminus in 26S Integrity

**DOI:** 10.3390/biom13060992

**Published:** 2023-06-15

**Authors:** Shirel Steinberger, Julia Adler, Yosef Shaul

**Affiliations:** Department of Molecular Genetics, Weizmann Institute of Science, Rehovot 76100, Israel; shirel.topaz@weizmann.ac.il (S.S.); julia.adler@weizmann.ac.il (J.A.)

**Keywords:** tagging 20S proteasome, tagging 19S regulatory particle, proteasome subcellular localization, proteasome granules, PSMA3 (α7)

## Abstract

Proteasomes critically regulate proteostasis via protein degradation. Proteasomes are multi-subunit complexes composed of the 20S proteolytic core particle (20S CP) that, in association with one or two 19S regulatory particles (19S RPs), generates the 26S proteasome, which is the major proteasomal complex in cells. Native gel protocols are used to investigate the 26S/20S ratio. However, a simple method for detecting these proteasome complexes in cells is missing. To this end, using CRISPR technology, we YFP-tagged the endogenous PSMB6 (β1) gene, a 20S CP subunit, and co-tagged endogenous PSMD6 (Rpn7), a 19S RP subunit, with the mScarlet fluorescent protein. We observed the colocalization of the YFP and mScarlet fluorescent proteins in the cells, with higher nuclear accumulation. Nuclear proteasomal granules are formed under osmotic stress, and all were positive for YFP and mScarlet. Previously, we have reported that PSMD1 knockdown, one of the 19 RP subunits, gives rise to a high level of “free” 20S CPs. Intriguingly, under this condition, the 20S-YFP remained nuclear, whereas the PSMD6-mScarlet was mostly in cytoplasm, demonstrating the distinct subcellular distribution of uncapped 20S CPs. Lately, we have shown that the PSMA3 (α7) C-terminus, a 20S CP subunit, binds multiple intrinsically disordered proteins (IDPs). Remarkably, the truncation of the PSMA3 C-terminus is phenotypically reminiscent of PSMD1 knockdown. These data suggest that the PSMA3 C-terminal region is critical for 26S proteasome integrity.

## 1. Introduction

We have a good knowledge of the 26S proteasome structure [1,2,3]. The 26S proteasome is formed by assembling two different protein complexes, a 20S catalytic core particle (20S CP), and the 19S regulatory particle (19S RP). Each of these particles, assembled independently but in the context of the 26S proteasomes, are subjected to mutual allosteric regulation [4]. In addition to the 19S RP, alternative complexes including PA200 (PSME4) and 11S/PA28 are associated with 20S CPs [5]. The 20S CP is built from two copies of seven different alpha (PSMA1 to 7) and seven different beta (PSMB1 to 7) subunits, each organized to form a ring shape. The two interior beta rings are flanked by the alpha rings in forming a cylindrical chamber [3]. The beta rings are endowed with proteolytic cleavage specificities, categorized as trypsin-like (PSMB7), chymotrypsin-like (PSMB5), and caspase-like (PSMB6) [2]. The PSMA flexible N-terminal regions form the gate that controls the access of substrates to the chamber [3,6]. The PSMA C-terminal flexible helices stick out of the CP and might interact with certain client proteins [7,8,9].

The multi-subunit 19S RP caps one or both ends of the 20S CP. The 19S RP docking is ATP-dependent, but can be substituted with NADH [10]. In the latter case, there is a preference for the degradation of intrinsically disordered proteins [11]. The 19S RP recognizes ubiquitinated substrates, is active in their deubiquitination, and enhances substrate accessibility into the 20S CP in an ATP-hydrolysis-dependent manner. The 19S RP is further subdivided into the base and lid subcomplexes [12,13]. Several proteasome-dedicated chaperones regulate 26S proteasome assembly [14] and activation [15]. The knockdown of PSMD1 19S RP subunits gives rise to the accumulation of 20S CPs that are either free or in complex with some alternative caps [16,17,18].

Proteasomes are localized in the cytoplasm and the nuclei of animal cells. The difficulty in determining the proteasome subcellular localization emerged from the tendency of proteasomes to reprogram their subcellular localization in response to growth conditions and a hostile environment [19,20]. In response to osmotic stress, specific nuclear proteasomal condensates, containing active proteasomes and polyubiquitinated proteins, have recently been described [21]. These nuclear proteasomal condensates or granules exhibit characteristics of liquid–liquid phase separation.

Several methods have been developed to visualize proteasomes in animal cells, such as the utilization of antibodies for immunofluorescent assay, or the overexpression of fluorescently tagged proteasome subunits [22]. Lately, mostly thanks to the CRISPR technology, the endogenous proteasome subunit can be fluorescently tagged. We have reported the generation of PSMB6-YFP, one of the beta ring subunits [17,23]. PSMB2-eGFP or PSMB2-FusionRed-KI HCT116 cells have been generated and used for time-lapse imaging of the 20S CP in the cells [21]. This group also employed PSMD6-eGFP to visualize the 19S RP. For 26S proteasome visualization, the cells need to be tagged at a subunit of both the 20S CP and the 19S RP. This is critical to monitoring the 26S/20S ratio. We established double-tagged HeLa cells co-expressing PSMB6-YFP and PSMD6-mScarlet to time-lapse imaging of the 20S and 19S particles at the single-cell resolution. We found that the vast majority of the 20S CPs were 19S-RP-capped.

The proteasome granules formed in the nuclei in response to osmotic stress were all YFP- and mScarlet-positive, suggesting that they were made of 26S proteasome. Interestingly, under PSMD1 knockout, the “free” 20S CPs remained nuclear, whereas the 19S RP was accumulated in the cytoplasm. The subcellular segregation of the two major proteasome components was also observed when the PSMA3 C-terminus was truncated, suggesting that the 20S CP regulated its capping. The double-tagged proteasome cell line was chosen for investigating 26S proteasome assembly in animal cells.

## 2. Materials and Methods

### 2.1. Cell Culture

The generation of the HEK293 PSMB6-YFP cell line was previously reported by us [17,23]. The G3BP1-mCherry construct was a gift from E. Hornstein, and originated by N. Kedersha and P. Anderson (Harvard Medical School, USA). The pFIRES puro plasmid was a gift from C. Kahana, and originated by S. Hobbs [24].

The cells were grown at 37 °C in a humidified incubator with 5% CO_2_ in high-glucose Dulbecco’s Modified Eagle Medium (DMEM; GIBCO, Life Technologies, Thermo Scientific, Waltham, MA, USA) supplemented with 8% fetal bovine serum (GIBCO, Waltham, MA, USA), and 100 units/mL penicillin, and 100 μg/mL streptomycin (Biological Industries, Beit Haemek, Israel).

### 2.2. Materials

Paraformaldehyde and poly-D-lysine hydrobromide, polybrene, and sodium (meta) arsenite were purchased from Sigma-Aldrich (Burlington, MA, USA); jetOPTIMUS^®^ was purchased from Polyplus (New York, NY, USA). PEI MAX^®^ was purchased from Polysciences (Warrington, PA, USA). NEBuilder^®^ HiFi DNA Assembly Cloning Kit was purchased from BioLabs (Ipswich, MA, USA). Hoechst was purchased from Molecular Probes (Eugene, OE, USA).

### 2.3. Immunoblot

Immunoblots were performed as previously described [25], using RIPA buffer (50 mM Tris-HCl pH 7.5, 150mM NaCl, 1% Nonidet P-40 (*v*/*v*), 0.5% deoxycholate (*v*/*v*), 0.1% SDS (*w*/*v*)) supplemented with cocktails of protease inhibitors (Apex Bio, Boston, MA, USA). The antibodies used were monoclonal anti-β-actin, polyclonal anti-PSMD6 (ab155761, Abcam, Cambridge, UK), monoclonal anti-RFPs (6G6, ChromoTek, Planegg, Germany), and the polyclonal Living Colors antibody (Clontech, Mountain View, CA, USA) (to detect YFP). The detection by Living Colors antibody and by psmd6 antibody was more sensitive if the samples were not boiled before separation by SDS-PAGE.

### 2.4. Lentivector Preparation and Transduction

Lentiviruses were produced as described [26] for the expression of G3BP-mCherry. For transduction, a medium containing lentivirions was filtered through a 0.45 µM filter, and supplemented with polybrene (8 μg/mL), and added to the cells for ~16 h. The transduced cells were washed with warm PBS three times, and a fresh medium containing 10 µg/mL Blasticidin was added to the cells for selection. After five days, the cells were sorted with FACSAriaIII (BD Biosciences, San Jose, CA, USA) according to the mCherry fluorescence intensity.

### 2.5. Establishment of Tagged-Proteasome Cell Lines

The establishment of the heterozygous HeLa and HEK293 PSMB6-SYFP edited cell lines was previously reported [17,27]. To establish cell lines expressing tagged 19S RPs, for expressing both the 20S CP and the 19S RP tagged proteins, we fused mScarlet to the C-terminus of PSMD6 in the HeLa cell line. YFP and mScarlet were used because they are not dimerized [28,29,30]. The SpCas9 expression plasmid was pX330-U6-Chimeric_BB-CBh-hSpCas9 (Addgene plasmid #42230; RRID: Addgene_42230). The guide sequence CACCGGCTTTACATATTAATTACTC was inserted into this plasmid in targeting the 3′ end of the PSMD6 gene. The donor plasmid was constructed by cloning a 1033 bp long fragment (Appendix A) of the 3′ end of the PSMD6 gene, followed by a linker sequence in pBlueScript KS- (Stratagene). The linker sequence GGTGGAGGTAGTGGGGATCCACCGGTCGCCACC was followed by a segment of the mScarlet coding frame and by a segment of 964 bp of the 3′ end of the human genome gene PSMD6 (from the stop codon in the reading frame).

The SpCas9 expression plasmid and the donor plasmids were transfected to HeLa cells using the jetOPTIMUS reagent, according to the company instructions. As the editing efficiency was extremely low (~1%), cells were also transfected simultaneously with pFIRES puro plasmid [24] for puromycin selection. Cells were selected using 2 µg/mL puromycin the day after transfection. After a further 14 days, cells were sorted using FACSAriaIII (BD Biosciences, San Jose, CA, USA) into cells expressing PSMD6-mScarlet according to the mScarlet fluorescence signal. In establishing the double-tagged reporter, HeLa PSMD6-mScarlet PSMB6-YFP cells, the PSMB6-YFP editing was conducted in the PSMD6-mScarlet cell line. Double-reported positive cells were sorted using FACS. The cells were heterozygous for PSMD6 (Appendix A).

### 2.6. Osmotic Stress Protocol

About 30,000 cells were plated in 96-well glass-bottom plates. A day later, cells were Hoechst-stained (5 µL/ml), and 20–30 min later, time-lapse images were taken. The media was replaced at time zero with media supplemented with 150 mM NaCl/200 mM sucrose for inducing osmotic stress.

### 2.7. Generation of PSMD1-KD Cells

Cells were generated as described by [17]. Briefly, cells were transduced with a lentiviral Tet-inducible TRIPZ vector with shRNAmir against 26S proteasome subunit PSMD1 (5′TGCTGTTGACAGTGAGCGAG**CTCATATT GGGAATGCTTA**TTAGTGAAGCCACAGATGTAATAAGCATTCCCAATATGAGCCTGCCTACTGCCTCGGA-3′). The PSMD1 sequence is bolded. Cells are viable under this condition, but die 4–5 days after dox treatment. Images were often taken at day three.

### 2.8. Generation of PSMA3 C-Terminal Truncated Cells

To truncate the PSMA3 C-terminal region, a guide targeting PSMA3 around the amino acid 203 was cloned into a single lentiviral vector for the delivery of Cas9, a sgRNA, and a puromycin selection marker (lentiCRISPRv2, #52961 from Addgene, Watertown, MA, USA). For the control, we used lentiCRISPRv2 harboring non-targeting gRNA sequences (TTTCGTGCCGATGTAACAC). The sgRNAs were designed, and off-target cutting was assessed, using the CRISPR design tool, Zhang lab, MIT. The lentivirions were produced as described [26]. The HeLa PSMD6-mScarlet PSMB6-YFP cells were transduced with lentivirions based on the above vectors, followed by selection with 2 µg/mL puromycin for 3 days, and were imaged the next day. The edited cells did not survive, and died 1–2 days later (Appendix A).

To prepare the genomic DNA for PCR analysis, cell pellets from approximately 10^5^ cells were suspended in 18 μL 50 mM NaOH, and heated at 95 °C for 10 min, followed by neutralization with 2 μL 1 M Tris-Cl pH 8. The PSMA3 genomic region around aa 203 was amplified, and the PCR fragment was sequenced via Sanger sequencing, using a 3730 DNA Analyzer (ABI). The Synthego ICE tool (https://ice.synthego.com/#/; accessed on 1 April 2021) was used to deconvolute the Sanger sequencing results of the edited clones.

### 2.9. In-Gel Analysis of the Fluorescently Tagged Proteins

SDS-PAGE was used to assess the expression of PSMB6-YFP and PSMD6-mScarlet subunits using an in-gel fluorescence assay [31]. HeLa PSMD6-mScarlet PSMB6-YFP cells were lysed on ice in NP-40 buffer [20 mM Tris-HCl (pH 7.5), 0.32 M sucrose, 5 mM MgCl_2_, 1% NP-40, 2 mM ATP, and 1 mM DTT, supplemented by protease inhibitor cocktail] for 20 min, following centrifugation (20 min at 27,650× *g* at 4 °C). Supernatants (40 μg) were mixed with the Laemmli protein sample buffer, resulting in the final 0.8% SDS, but were not heated, and were separated on 12.5% SDS-PAGE. Gels were then scanned by the Typhoon FLA 9500 imager for the YFP signal, using a 473 nm laser for excitation, and a 530DF20 BPB1 emission filter; and mScarlet signal, using the 532 nm laser for excitation, and 575LPR (>665 nm emission filter).

To visualize the proteasomal complexes in native conditions, the above extracts (80 μg) were analyzed on a nondenaturing 4% polyacrylamide gel, and the fluorescent signals were detected as above.

### 2.10. Live Cell Imaging

Cells were seeded on a 96-well glass bottom Microwell plate 630 µL black 17 mm low glass from Matrical Bioscience (MGB096-1-2-LG), and were allowed to adhere overnight. HeLa was at a density of 30,000 cells/well, and HEK293 was at a density of 200,000 cells/well, in a well that was coated with poly-D-lysine hydrobromide. Before the imaging using spinning disk microscopy, the cell media were supplemented with 5 µg/mL Hoechst 33342. The plate was placed in the microscope chamber, and cells were maintained at 37 °C and 5% CO_2_ for the duration of the experiment. Images of four different sites in a well were taken every 4 min for 3 h. Images were taken using a VisiScope Confocal Cell Explorer live cell imaging system with a 60-oil objective.

### 2.11. Data Processing and Analysis

The images were constructed and processed using Visiview, Imaris, and Excel software. The images were cropped and adjusted using Visiview software. The detection of nuclei and granules was conducted using a different set of parameters for every experiment in Imaris. Nuclei were defined as surfaces; the threshold was 400–1000 (absolute intensity). Granules were defined as spots; the threshold was 100–115 for YFP and 45–60 for mScarlet. The classification of granules inside/outside nuclei was calculated using the shortest distance to surfaces, a distance between −2 and −6 µM. For mScarlet intensity analysis, nuclei and cytoplasm detection was conducted using the definitions of cells. Nuclei were defined using the Hoechst marker; the threshold was 350–400. Cytoplasm was defined using the mScarlet marker; the threshold was 250–300. The data were then processed, and presented in graphs in Excel. The profile line plot was generated by first manually drawing a linear region of interest, and exporting the value along the line for each channel. The FIJI ROI Manager tool’s multi-plot feature was used for this step [32]. Next, the values from each channel were normalized to that channel’s respective maximum value, and the normalized values were plotted in Excel.

Statistical tests of the two-tailed *t*-test were performed to assess significance, using Excel formulae. In the case of the mScarlet intensity level, a 100 background count was reduced from the total amount of both nuclei and cytoplasm.

### 2.12. Proteasome Modeling

Proteasome modeling was conducted using the program PyMOL. PDB files were downloaded from the RCSB Protein Data Bank (RCSB PDB) website. Files were: 5vhh.PDB, 7oin.PDB, 3ed8.pdb, 6rgq.pdb.

## 3. Results

### 3.1. Intracellular Monitoring of the 26S Proteasome

We have previously reported the establishing of a YFP-tagged 20S proteasome cell line [17,23]. To this end, we employed CRISPR technology to generate the chimeric PSMB6-YFP gene. We transduced the PSMB6-YFP cell line using a lentivector-expressing G3BP1-mCherry, a reporter of the stress granules. The tagged proteasome was preferentially nuclear, whereas the G3BP1-mCherry was exclusively in the cytoplasm (Figure 1A). We exposed the cells to osmotic stress, to induce the formation of nuclear proteasomal granules (Figure 1B). Next, we treated the cells with arsenite, an inducer of stress granules. Stress granules were selectively formed under arsenite treatment. These data validate the distinct behavior of the proteasomal granules from the stress granules.

Next, we modified the PSMB6-YFP cells by fusing the endogenous PSMD6 gene to the mScarlet fluorescent protein, using the CRISPR technique to establish a double-tagged proteasome in the HeLa cells (Figure 2A). The proteasomal activity of the established cell line was not affected (Appendix A). The expression of the fluorescent PSMB6 and PSMD6 proteins was analyzed using SDS-PAGE and native gel (Figure 2B). As expected, the 26S and the 20S proteasome complexes were both positive for PSMB6-YFP, whereas only 26S was positive for PSMD6-mScarlet. The 26S proteasome, therefore, was positive for both PSMB6-YFP and PSMD6-mScarlet. PSMD6-mScarlet allowed the detection of the intracellular 19S RP in real time.

Next, we microscopically analyzed the cells for the distribution of the fluorescent proteins. Interestingly, in the majority of the cells, an enhanced nuclear fluorescence signal could be observed (Figure 2C and Appendix A). The intracellular distribution of PSMB6-YFP completely overlapped with the PSMD6-mScarlet distribution. Furthermore, the proteasome granules formed under osmotic stress were all positive for both YFP and mScarlet. These data suggest that the 26S proteasome is specifically visualized in the double-tagged cell line, and that the imaging does not easily distinguish between the other known proteasome complexes (free 20S CP, PA200, and 11S/PA28).

### 3.2. Time Kinetics of the 26S Proteasome Granules in Response to Osmotic Stress

Having established a cell line for visualizing the 26S proteasome, we next performed time-lapse imaging to monitor the behavior of the 26S proteasome under osmotic stress. Cells were treated with 200 mM sucrose and, as expected, described by [21], hyperosmotic stress induced the formation of nuclear PSMB6-YFP positive proteasome granules (Figure 3A and Appendix A). The 26S proteasomes formed these granules, as all the granules were also positive for the PSMD6-mScarlet. The 26S granules were visualized after four minutes, and reached maximal level after 12 min (Figure 3B,C). Cell recovery took place with slower time kinetics. Interestingly, at the longer time points, the granules were more visualized at the nuclear periphery.

Next, we induced salt osmotic stress (Figure 4A and Appendix A). Here again, all the granules were positive for both PSMB6-YFP and PSMD6-mScarlet, suggesting that the 26S proteasome formed the granules. Salt-osmotic-stress-induced proteasomal granules rapidly developed, and the maximal number of granules was observed after 4 min (Figure 4B,C). Similarly to the sucrose treatment, the decay in the number of granules took much longer, with a tendency to be localized at the nuclear periphery. These data suggest that in response to osmotic stress, the 26S proteasomes form nuclear granules, and that no 26S disassembly occurs along the formation and resolution of the granule formation.

### 3.3. Nuclear Localization of the 20S Proteasome in PSMD1 Knockdown Cells

Previously, we have shown that the knockdown of subunits of the 19S RC reduces the level of the 26S proteasome, with a concomitant increase in the level of the 20S CPs [17]. We transduced the proteasome double-reporter cells with a lentivector expressing shRNA to target PSMD1 in a doxycycline (dox) inducible manner (Figure 5A), using our reported protocol [17]. Under this condition, the level of most of the other 19S RP was not affected (Appendix A). Native gel analysis validated a sharp reduction at the level of the 26S, with a concomitant increase at the level of the 20S CP, in dox-treated cells (Figure 5B). PSMD1 (Rpn2) is a component of the 19S base subcomplex, and is critical to the assembly of the 19S RP [12]; indeed, the level of the intact 19S RP is too low to be detected. In fact, it is expected that the 19S RP is completely disassembled, and PSMD6-mScarlet is “free”. As PSMD6-mScarlet is a component of the 19S lid, the observed faint red complex underneath the 20S CP is likely to be the residual PSMD6-mScarlet positive lid subcomplex. Remarkably, in dox-treated cells, the PSMD6-mScarlet was barely detected in the nuclei, and significantly accumulated in the cytoplasm (Figure 5C,D). The cytoplasmic PSMD6-mScarlet is likely to represent the “free” or 19S-RP-lid-associated PSMD6-mScarlet. The “free” 20S CPs probably remained mostly nuclear. This behavior was not observed in the control cells; the dox-treated cells in the absence of the PSMD1 shRNA cassette, and the non-treated cells in the presence of the PSMD1 shRNA cassette. These results suggest that the “free” 20S CPs are probably mostly nuclear.

Next, the cells were subjected to osmotic stress (Figure 5E and Appendix A). As expected, nuclear 26S proteasome granules were rapidly formed in untreated control cells [21,33]. In contrast, in the PSMD1 knocked-down cells, the PSMD6-mScarlet remained in the cytoplasm and did not form visible granules. The “free” 20S CP remained nuclear, but did not form granules. Before osmotic stress induction, the 20S CP was homogeneously distributed in the nucleus; however, upon osmotic stress, the 20S CP did not form the classical granules, but instead formed subnuclear diffused aggregates. These data suggest that the 26S proteasome is critical to the formation of proteasome granules in response to osmotic stress.

### 3.4. PSMA3 C-Terminus Regulates 26S Proteasome Integrity

Lately, we have reported that the PSMA3 C-terminus (187–255 amino-acids) that we termed “trapper” interacts with many disordered proteins in their degradation [9]. We employed CRISPR technology to truncate the PSMA3 C-terminal region in the proteasome double-reporter cells, and the pooled cells were subjected to further analysis (Figure 6A,B). For a technical reason, the truncation was conducted on the 203 residue that deletes the most of the trapper region. As a control, we used the lentiCRISPRv2 harboring non-targeting gRNA sequence. “Free” 20S CPs were not detected in the C-terminus truncated PSMA3 cells, as analyzed by native gel (Figure 6C). Although RNA analysis revealed that the cells express about 20% truncated PSMA3 mRNA, we could not detect the truncated protein by immunoblot analysis. Moreover, the 19S RP became unstable, and only the lid subcomplex was detected. Thus, the proteasome integrity was dramatically compromised under these conditions. The PSMB6-YFP was nuclear, and the PSMD6-mScarlet was localized in the cytoplasm, including the residual 19S RP (Figure 6D,E). For controls, we either used irrelevant gRNA, or transduced the cells with an “empty” lentivector.

Next, the cells were subjected to osmotic stress (Figure 6F). The observed phenotype was reminiscent of the PSMD1 knocked-down cells, and the nuclear PSMB6-YFP did not form foci. The few observed foci were all positive to PSMD6-mScarlet, suggesting that they were of the 26S proteasome type. These data suggest that the CRISPR editing targeted to delete the C-terminal region of the PSMA3 dramatically affected the integrity and subcellular localization of both the 20S CP and the 19S RP.

## 4. Discussion

We report here the establishment of a cell line for the live visualization of both the 20S CP and the 19S RP at a single-cell resolution. To this end, we utilized YFP and mScarlet fluorescent tags that are unlikely to heterodimerize or to generate false colocalization signals [28,29,30]. Native gel analysis and time-lapse imaging revealed that these two complexes form the 26S proteasome, under normal and osmotic stress conditions. However, the data should be cautiously treated, as the fluorescence tag has a signal radius that can overlap with molecules which are not near the tagged object. The proteasomes appear to also reside in the nucleus, and the osmotic stress-induced proteasome granules are primarily nuclear. The alternative 20S CP caps, such as PSME1 to 4 [5], which in our assay should be tagged only with PSMB6-YFP, the “free” 20S CPs, are hardly seen. Using biochemical and extract fractionation methods, it has been reported that “free” 20S CPs are about 30–40% of the total proteasome complexes in HeLa cells [34,35]. We did not visualize a significant amount of “free” 20S CPs, signifying the meager amount of non-26S proteasomes. However, one cannot exclude the possibility that certain levels of “free” 19S and 20S are co-localized without forming 26S. Notably, asymmetric 26S proteasomes, where the 20S CP is capped with only one 19S RP, with the antipode side being uncapped (RP-CP, but not RP-CP-RP), cannot be distinguished by our assay from those that are doubly capped, as a single RP is sufficient to label the 20S CP with mScarlet. The unoccupied side of the 20S CP (RP-CP) might function in a 19S RP-independent manner.

The knockdown of a 19S RP subunit increases the level of the “free” 20S CPs [16,17,18], but whether this process affects the proteasomal subcellular localization is unknown. We report here that under this condition, the tagged 19S subunit, possibly the 19S RP lid subcomplex, is nuclear-excluded, whereas the “free” 20S CPs are mostly nuclear. The “free” 20S CPs are nuclear even under osmotic stress, although they do not form the classical condensate granules of the 26S proteasome. It has been reported that RAD23B, a ubiquitin-binding shuttle protein, is essential in the formation of nuclear proteasomal condensates [21,33,36]. The p62/SQSTM1 ubiquitin-binding protein is another ubiquitin shuttle protein regulating nuclear condensates [33,37]. The nuclear “free” 20S CPs could be either the result of their de novo nuclear assembly, or the product of the 26S proteasomes’ disassembly. However, since this proteasome complex is not expected to bind ubiquitin shuttle proteins, it seems that their nuclear residency is either the default of 26S proteasome disassembly, or is AKIRIN2-mediated. AKIRIN2 binds to the 20S proteasome surface in mediating its nuclear import of the “free” 20S CPs [38,39,40].

Evidence is accumulating for the role of the 20S CP in isolation in degrading proteins [9,16,41,42]. Since the process is ubiquitin-independent, a key question was how the substrates are recruited for degradation. In vitro degradation assays have revealed over two hundred IDP/IDR proteins that are degraded by purified 20S CPs [43]. Interestingly a large fraction of this group of IDPs/IDRs directly bind the PSMA3 C-terminus [9]. To investigate the role of this region in the cells, we employed the CRISPR technology to truncate PSMA3, by deleting the region active in binding IDP/IDR substrates. Using the proteasome double-reporters, we found a profound segregation of the two tagged proteasome subunits; the 20S CP PSMB6-YFP was nuclear, whereas the 19S RP PSMD6-mScarlet subunit resided in the cytoplasm. This finding is consistent with the report that PR-39 blocked the PSMA3 (alpha 7) subunit and inhibited 20S-19S interaction [44]. Native gel analysis has revealed that the tagged 19S RP is not intact, and is possibly dissociated into its two subcomplexes, base and lid [12]. Under osmotic stress, the tagged 19S RP remained in the cytoplasm. The nature of the nuclear PSMB6-YFP-positive component is not well defined. The beyond-detection level of the 20S CP in the native gel, and lack of detection of the truncated PSMA3 might suggest that the observed nuclear PSMB6-YFP is not an intact 20S CP. However, it is also possible that the incorporation of the truncated PSMA3 into the 20S CP makes the complex unstable, and dissociates during cell extraction.

How PSMA3 regulates 26S proteasome integrity is an open question. Previously, it has been reported that PSMA3 C-terminal phosphorylation increases 26S integrity [45]. This region has an acidic patch, and phosphorylation further increases acidity. It is, therefore, possible that the charged PSMA3 C-terminus increases the process of 20S CP capping by 19S RP, as was reported in yeast for the binding of Ecm29, a proteasome interacting protein [7,8]. The double-reporter proteasome cell line we have established, for the first time, allows the live visualization of proteasome distribution in mammalian cells, to investigate proteasome behavior under various conditions. In isolation, the distinct subcellular localization of the two major proteasome particles, the 20S CP and the 19S RP, can be exploited to investigate 26S proteasome assembly, and identify the proteins involved in this process. Furthermore, this system might greatly assist in discovering and assaying small molecules regulating proteasome assembly and disassembly, for research and clinical aims [17,46,47].

## Figures and Tables

**Figure 1 biomolecules-13-00992-f001:**
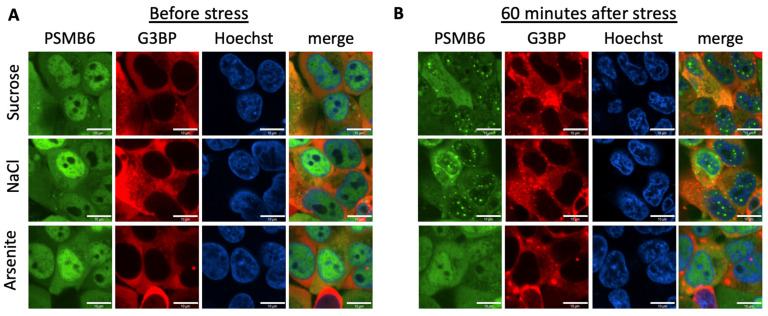
PSMB6 and G3BP under different stresses; the positive proteasome bodies are distinct from the stress granules. (**A**) PSMB6 was tagged with YFP, and G3BP1 was tagged with mCherry in HEK293 cells. (**B**) Cells were subjected to treatment with arsenite (1 mM, 1 h), sucrose (200 mM, 1 h), or NaCl (150 mM, 1 h). The green live-imaging marker represents the 20S; the red represents stress granules. Nuclei were stained using Hoechst staining (blue). The scale bar is 10 µm.

**Figure 2 biomolecules-13-00992-f002:**
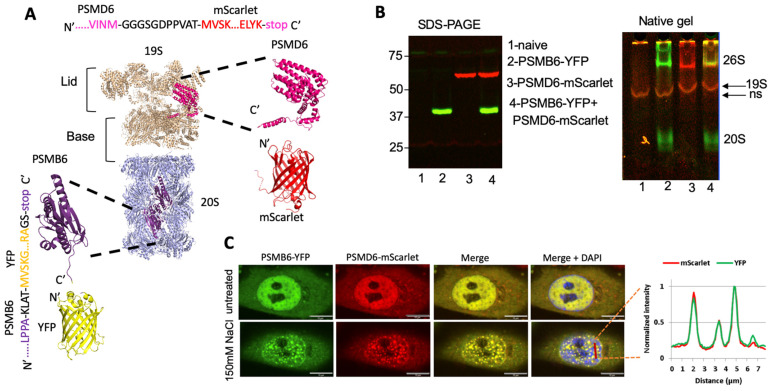
Establishment of a double-reporter HeLa cell line. (**A**) Model showing the chimeric proteasome proteins and the fluorescence proteins. The 20S PC core proteasome is in gray, the PSMB6 subunits in purple, the YFP fluorescence protein in yellow, the 19S RP in cream, the PSMD6 subunit in pink, and the mScarlet fluorescence protein in red. The junction sequences, PSMD6 C-terminus, and mScarlet N-terminus, are shown. (**B**) Fluorescent detection of the chimeric proteins. On the left is SDS-PAGE, and on the right is the native gel analysis of the cell extracts. 19S is visualized but, very close to it, a nonspecific (ns) band is seen in the controls. The identity of the 19S becomes more evident; see below. (**C**) HeLa PSMB6-YFP PSMD6-mScarlet cells were treated with 150 mM NaCl. The green live-imaging marker represents the 20S, the red marker 19S, and hence 26S; the blue represents the nucleus. The scale bar is 10 µm. The left panel shows the line profiling of a section of the cell, indicated by a red dashed line.

**Figure 3 biomolecules-13-00992-f003:**
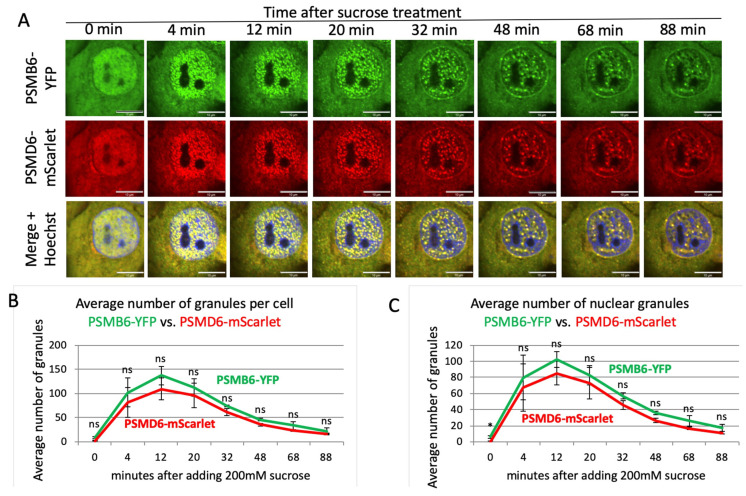
Proteasome granule dynamic under sucrose osmotic stress. (**A**) Time-lapse images of HeLa PSMB6-YFP PSMD6-mScarlet cells treated with 200 mM sucrose. The green live-imaging marker represents all proteasomes, the red live-imaging marker represents 19S RP, and the yellow spots in the merge panel represent 26S proteasomes. The nuclei were Hoechst-stained (blue). Frames were taken every 4 min. The scale bar is 10 µm. (**B**) The average number of granules per cell at the indicated time points seen with YFP vs. mScarlet markers; *n* = 3 repetitions. (**C**) The average number of nuclear granules at the indicated time points seen with YFP vs. mScarlet; *n* = 3 repetitions. ns is not significant, * *p* < 0.05 by two-tailed unpaired *t*-test.

**Figure 4 biomolecules-13-00992-f004:**
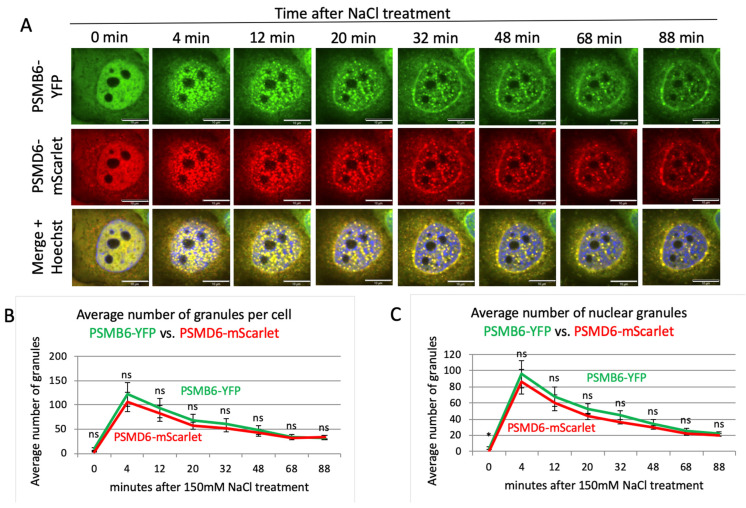
Proteasome granule dynamic under NaCl osmotic stress. (**A**) Time-lapse images of HeLa PSMB6-YFP PSMD6-mScarlet cells treated with 150 mM NaCl. The green live-imaging marker represents all proteasomes, the red live-imaging marker represents 19S RP, and the yellow spots in the merge panel represent 26S proteasomes. The nuclei were Hoechst-stained (blue). Frames were taken every 4 min. The scale bar is 10 µm. (**B**) The average number of granules per cell at selected time points seen with YFP vs. mScarlet markers; *n* = 3 repetitions. (**C**) The average number of nuclear granules at the selected time points seen with YFP vs. mScarlet; *n* = 3 repetitions. ns is not significant., * *p* < 0.05 by two-tailed unpaired *t*-test.

**Figure 5 biomolecules-13-00992-f005:**
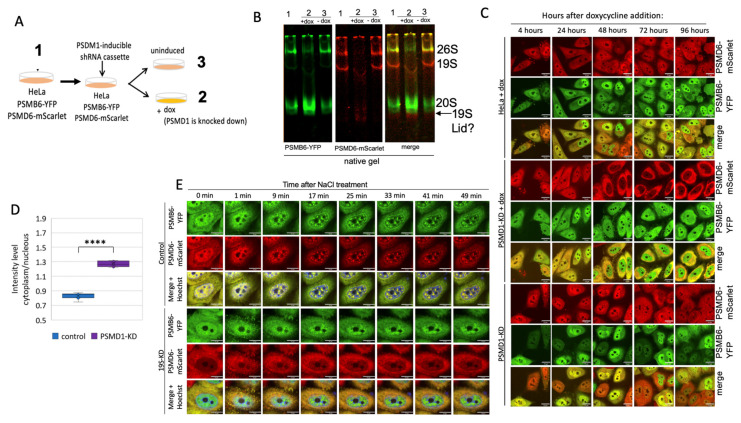
Proteasome granules are dynamic under osmotic stress. (**A**) Schematic description of the strategy of PSMD1 knockdown using the doxycycline (dox) inducible system. (**B**) Native gel of the three treatment groups described in panel (**A**), as scanned for YFP and mScarlet. (**C**) HeLa PSMB6-YFP PSMD6-mScarlet PSMD1-KD cassette cells were treated or untreated with doxycycline and imaged after 3.5 days. As an additional control, HeLa PSMB6-YFP PSMD6-mScarlet cells were doxycycline-treated and imaged after 3.5 days. (**D**) Boxplot analysis of the mScarlet intensity level in the nucleus versus the cytoplasm, *n* = 8 repetitions each. **** *p* < 0.0001 by two-tailed unpaired *t*-test. (**E**) Time-lapse images of HeLa PSMB6-YFP PSMD6-mScarlet cells, and HeLa PSMB6-YFP PSMD6-mScarlet PSMD1-KD cells, treated with 100 mM NaCl. Frames were taken every 4 min. The scale bar is 10 µm.

**Figure 6 biomolecules-13-00992-f006:**
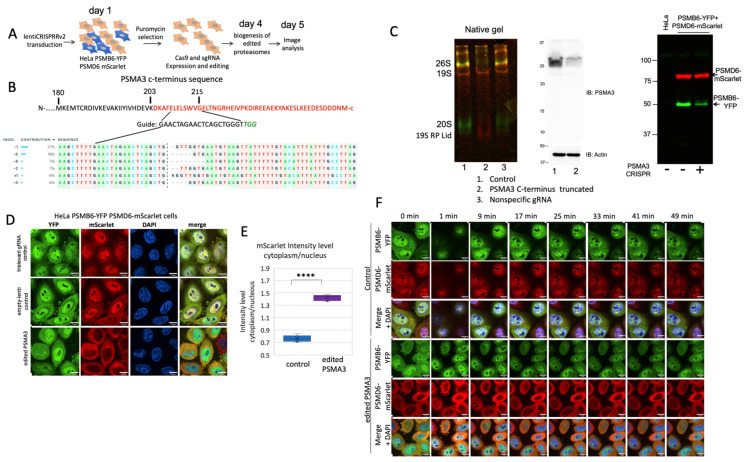
The PSMA3 C-terminus regulates 26S proteasome integrity. (**A**) Schematic description of the strategy of selecting cells with truncated C-terminus PSMA3. A day after transduction, cells were puromycin-treated (2 µg/mL) for three days. Images were taken a day later. (**B**) Sequencing analysis of edited HeLa PSMB6-YFP PSMD6-mScarlet cells in deleting the PSMA3 C-terminal region. The 450 bp fragment was amplified and sequenced, and the results were analyzed using the Synthego ICE tool (Synthego Performance Analysis, ICE Analysis. 2019. v2. Synthego). The guide sequence is shown. (**C**) Native, immunoblot, and denaturating gel analysis of the CRISPR edited cells. The PSMA3 monoclonal antibody used was D4Y9O (cell signaling). (**D**) The edited HeLa PSMB6-YFP PSMD6-mScarlet cells and the controls (lentiviruses with a non-targeting guide or with a trapper-targeting guide) were imaged after 5 days. The scale bar is 10 µm. (**E**) Boxplot analysis of the mScarlet intensity level in the nucleus versus the cytoplasm. **** *p* < 0.0001 by two-tailed unpaired *t*-test. (**F**) Time-lapse images of cells treated with 100 mM NaCl. Frames were taken every 4 min. The scale bar is 10 µm.

## Data Availability

Not applicable.

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
