# Peer review of "Method of Monitoring 26S Proteasome in Cells Revealed the Crucial Role of PSMA3 C-Terminus in 26S Integrity"

_biomolecules, 2023, doi:10.3390/biom13060992_

Round 1
Reviewer 1 Report
Steinberger et al describe an approach to label proteasome CP and RP in order to visualize their co-localization in cells. From this effort they make several observations and some new claims. The authors use a variety of appropriate approaches to evaluate proteasome complexes and localization. The authors also use quantification that helps to clarify their observations. Overall, this study provides details about several new tools to study the proteasome in cells. These tools have utility, as presented in this study, and will be a great resource for the proteasome field. I have some concerns with the conclusions made based on the findings presented.
1) It is unclear whether the tagged proteasome complexes are catalytically active. The authors should at least do plate reader AMC assays or some form of activity based probe. The authors use native gels and thus could show in-gel AMC assays. Anything to confirm that the tags do not interfere with activity would be of value to the study. The reason for this being important is that inactive proteasomes may localize differently.
2)The authors should consider that fluorescent tags can dimerize. While it is likely that the authors interpretation of the images may be correct (i.e. "....vast majority of the 20S CP are 19S RP capped.") it is also possible that some fraction of the data is a result of this caveat. The authors should address this in the text as a possibility.
3) The authors are using fluorescent microscopy to make the claim that these single proteasome (i.e. CP and RP) particles are co-localizing. It is the case that at the resolution being used such a conclusion can not be made definitively. The fluorescence tag has a radius of signal which can overlap with molecules which are not near the tagged object. Hence the diffuse labeling. This is the purpose for higher resolution imaging such as light-sheet microscopy. Alternatively, single molecule imaging would go a long way to improve these conclusions.
4) The authors say that the localization is "preferentially nuclear" without quantification. From the images it is clear that there is also a lot of diffuse signal in the cytosol. Because the signal is so diffuse for both the CP and RP it is hard to tell whether this is co-localization or just overlap of fluorescence signal. The authors should include this line of thinking when describing their data.
5) I have a hard time accepting this statement given the caveats listed above. "These data suggest that the 26S proteasome is specifically visualized in the double-tagged cell line and that the level of other proteasome complexes (free 20S CP, PA200, and 11S PA28) is too low under detection by our technique." The native gels even show a significant amount of non-19S capped 20S CP. I would change this statement to say that the imaging does not easily distinguish between the many known subcomplexes of the proteasome.
6) Again, the statement "These results suggest that the “free” 20S CP is mostly nuclear" is hard to appreciate given the limitations of the approach. This statement, and others like it, should be changed to reflect the limitation of the imaging approach.
7) I find it difficult to understand figure 6. The authors should provide a denaturing gel of these samples to show that there is expressed tagged CP and RP subunits when PSMA3 is mutated. I am not clear how the authors see any YFP signal in cells when there is very little signal on the native gel.
Author Response
was uploaded

Reviewer 2 Report
The presented manuscript by Steinberger et al. reports endogenous and simultaneous CRISP/Cas-mediated tagging of both 20S and 19S proteasome subunits aimed at to define their sub-cellular localization under normal and stress conditions. In my opinion, this is interesting and solid study performed at a high technical level. However, it requires several points of critics should be addressed prior to publication in Biomolecules:
1. Lines 32-33 and 241-242: the alternative proteasomes regulator is known either as 11S or as PA28, but not as 11S PA28. The authors should put dash symbol in between these alternative names.
2. Line 40: a note on functional significance of C-terminus of proteasomes’ alpha-subunits should be added.
3. Line 41: authors must be aware that 19S proteasomes regulator does not exclusively consist of PSMD subunit, but also includes PSMC subunit. The phrase “the multi-PSMD subunit 19S RP” should be rewritten as “The multi-subunit 19S RP”
4. Line 49: it should be clarified which 19S RP subunit knockdown leads to the accumulation of free 20S CP, especially because, according to the references in the manuscript, there are three of them: PSMD1, PSMD2, and PSMD11
5. Line 229: the sentence “PSMD6-mScarlet allows the detection of the intracellular 19S RP in real time” should be put at the end of the paragraph instead (line 234).
6. Figure 2: the fluorescence of a single cell cannot be assessed, especially when it is explicitly stated in the paper that the cell tagging was heterozygous and, according to Supplementary materials (Figures S4 and S5), different distribution of fluorescence in different cells can be clearly seen. A micro-photograph of multiple cells must be presented in this Figure to show general state of the cell population in the experiment.
7. Line 236: it is argued that “Interestingly, the most fluorescent signal is localized in the nucleus (fig 2C)”. My previous comment is focused on how Supplementary materials (Figures S4 and S5) demonstrate various fluorescence distributions in different cells. Therefore, a preferred statement is that in the majority of cells, an enhanced nuclear fluorescence signal can be observed.
8. Lines 236 -241: authors insist that "the intracellular distribution of PSMB6-YFP is completely overlapped with PSMD6-mScarlet distribution", but under close inspection of the merge photo of the control cells, distinct areas of green and red fluorescence can clearly be seen. I agree that the line profiling of the chosen section of the cell under osmotic stress shows a good overlap of red and green signals, yet on a similar area in the control cells, there is no such overlap. Thus, a part of the sentence “the level of other proteasome complexes (free 20S CP, PA200, and 11S PA28) is too low under detection by our technique” (Lines 240-241) shall be removed, as the authors cannot estimate neither the number nor the distribution of these proteasome complexes in cells using their approach.
9. Chapter title “Nuclear localization of the 20S proteasome in 19S knockdown cells” (Line 292). The authors did not perform a 19S RP knockdown, thus “19S KD” should be changed to “PSMD1 KD”
10. Figure 5C: a merge panel shall be added (Red + Green fluorescence)
11. Lines 307-308 and line 390: in the statement "The "free" 20S CP remained mostly nuclear" the authors should add "probably" because they have not proved this statement; to do so they should perform other experiments, such as fluorescent photobleaching technique (for example, FLIP) or proteasome-activity probe staining of cells.
12. Paragraph (lines 293-311): did the PSMD1 knockdown have an effect on cell viability? If so, how long after Dox-treatment did this occur and what percentage of dead cells was there? Could the cells be rescued by transient re-expression of PSMD1?
13. Chapter "PSMA3 C-terminus regulates 26S proteasome integrity": it is not clear why the gRNA around the amino acid 203 was chosen. This should be explained at the beginning of the chapter.
14. Lines 342-344: I strongly disagree with the following statement “The PSMB6-YFP residual 20S CP is nuclear, possibly the result of some residual 20S CP and PSMD6-mScarlet is localized in the cytoplasm including the residual 19S RP”. According to the native assay, no fluorescent-labeled 19S, 20S, or 26S/30S complexes remain in the C-terminus-truncated PSMA3 cells, so it cannot be claimed that the green nuclear fluorescence in these cells represents 20S CP. The authors may rewrite it by "The PSMB6-YFP is nuclear and PSMD6-mScarlet is localized in the cytoplasm including the residual 19S RP”. Otherwise, nuclear localization of 20S CP shall be proven using methods such as FLIP, proteasome active probe cell staining, 4-20% native analysis, glycerol gradient, etc.
Author Response
is uploaded

Round 2
Reviewer 1 Report
Thank you.
Author Response
no comments
Reviewer 2 Report
Majority of manuscript shortcomings I raised in the initial review have been satisfactorily addressed. However, there is one correction that should be made for clarity.
1. Lines 32, 247: Use 11S (PA28) or 11S/PA28 abbreviation for this alternative proteasome regulator, not 11S PA28 or 11S-PA28
Author Response
Lines 32, 247: Use 11S (PA28) or 11S/PA28 abbreviation for this alternative proteasome regulator, not 11S PA28 or 11S-PA28.
done